# Preventing cation intermixing enables 50% quantum yield in sub-15 nm short-wave infrared-emitting rare-earth based core-shell nanocrystals

Fernando Arteaga Cardona[1], Noopur Jain[2,3], Radian Popescu[4], Dmitry Busko[1], Eduard Madirov[1], Bernardo A. Arús [5,6,7,8,9], Dagmar Gerthsen[4], Annick De Backer [2,3], Sara Bals [2,3], Oliver T. Bruns [5,6,7,8,9], Andriy Chmyrov [5,6,7,8,9] ✉, Sandra Van Aert [2,3] ✉, Bryce S. Richards [1,10] ✉ & Damien Hudry [1] ✉

Short-wave infrared (SWIR) fluorescence could become the new gold standard in optical imaging for biomedical applications due to important advantages such as lack of autofluorescence, weak photon absorption by blood and tissues, and reduced photon scattering coefficient. Therefore, contrary to the visible and NIR regions, tissues become translucent in the SWIR region. Nevertheless, the lack of bright and biocompatible probes is a key challenge that must be overcome to unlock the full potential of SWIR fluorescence. Although rare-earth-based core-shell nanocrystals appeared as promising SWIR probes, they suffer from limited photoluminescence quantum yield (PLQY). The lack of control over the atomic scale organization of such complex materials is one of the main barriers limiting their optical performance. Here, the growth of either homogeneous ($\alpha$-NaYF$_4$) or heterogeneous (CaF$_2$) shell domains on optically-active $\alpha$-NaYF$_4$:Yb:Er (with and without Ce$^{3+}$ co-doping) core nanocrystals is reported. The atomic scale organization can be controlled by preventing cation intermixing only in heterogeneous core-shell nanocrystals with a dramatic impact on the PLQY. The latter reached 50% at 60 mW/cm$^2$; one of the highest reported PLQY values for sub-15 nm nanocrystals. The most efficient nanocrystals were utilized for in vivo imaging above 1450 nm.

Over the last decade, short-wave infrared (SWIR) fluorescence (1000-2000 nm) has appeared as a promising route for realizing the next generation of imaging technologies in the biomedical field[1]. With significantly reduced scattering loss and near-zero auto-fluorescence, SWIR emitters can dramatically enhance spatial resolution, contrast, signal-to-noise ratio, and penetration depth. This can potentially enable important applications, such as real-time tracking of metabolic processes in living healthy/diseased tissues with single-cell sensitivity or improved clinical diagnosis and medical practice (e.g. preoperative imaging, image-guided surgery).

To date, various materials have been considered as potential probes for imaging in the SWIR region, including small organic molecules[2], single-walled carbon nanotubes[3], semiconductor quantum dots[4–6], and rare earth (RE)-based nanocrystals (NCs)[7–11]. Despite this,

A full list of affiliations appears at the end of the paper. ✉e-mail: andriy.chmyrov@nct-dresden.de; sandra.vanaert@uantwerpen.be; bryce.richards@kit.edu; damien.hudry@kit.edu

current SWIR materials still suffer from disadvantages – including low photoluminescence quantum yield (PLQY), toxicity concerns, and limited emission wavelengths – that can be crippling when considering biomedical applications.

RE-based NCs recently appeared as promising candidates for SWIR imaging, exhibiting no photobleaching, long luminescence lifetimes (e.g. for time-gated imaging), large Stokes shifts, integral coverage of the SWIR region with narrow emission bandwidths, and low long-term cytotoxicity[12–14]. Nevertheless, such materials suffer from relatively low PLQYs and very low absorption cross-sections, leading to poor brightness[1,15]. Thus, one of the major detrimental consequences is that RE-based NCs still have to be injected in large doses to achieve similar or better imaging results compared to other materials (e.g. indocyanine green or quantum dots)[4,15–17]. A further limitation for synthesizing highly efficient RE-based SWIR-emitting NCs originates from the limited understanding and lack of control over the real atomic-scale organization at the core-shell interface, thus leading to uncontrolled energy migration pathways[18]. Indeed, over the past few years, compelling experimental evidence for cation intermixing during the synthesis of targeted core-shell RE-based NCs were reported[19–23]. Such a phenomenon can dramatically modify intra-particle energy migration pathways with direct consequences for important optical characteristics such as the PLQY.

In this study, the atomic scale organization and optical performance of expected core-shell structures prepared from the exact same core NCs ($\alpha$-NaYF$_4$:Yb:Er without or with Ce$^{3+}$ co-doping) combined with either homogeneous ($\alpha$-NaYF$_4$) or heterogeneous (CaF$_2$) shell domains is investigated. In contrast to the utilization of a homogeneous shell domain that leads to severe cation intermixing (alloy formation), employing a heterogeneous shell domain leads to phase segregation (true core-shell structure). Such a massive refinement in the atomic scale organization of the as-synthesized nanostructures dramatically boosts the downshifting (980→1530 nm) PLQY, which reaches 50% for sub-15 nm NCs at 60 mW/cm$^2$. The latter is an order of magnitude lower compared to the skin exposure limit (730 mW/cm$^2$ at 980 nm) according to the guidelines from the International Commission on Non-Ionizing Radiation Protection[24]. Since SWIR imaging will be, in the near future, a cornerstone for preclinical and clinical imaging, the results reported here constitute a paradigm shift for the emergence of highly efficient RE-based SWIR emitting materials.

## Results

### Sub-10 nm optically active core nanocrystals: structural characterization

A single unique batch of cubic (Fig. 1a) $\alpha$-NaYF$_4$:Yb:Er (Y: 50 at.%, Yb: 45 at.%, Er: 5 at.%) core NCs was synthesized following a thermal decomposition method. High-angle annular dark-field scanning transmission electron microscopy (HAADF-STEM) confirms the formation of sub-10 nm isotropic particles with a spherical shape (Fig. 1b). The corresponding size distribution histogram reveals the co-existence of two populations with sizes of 6.0±1.2 nm and 9.2±1.0 nm (Fig. 1c). The crystal structure of the synthesized core NCs was checked by powder X-ray diffraction (PXRD). Rietveld refinement confirms the formation of the pure cubic phase with no trace of secondary phases (Fig. 1d). The refined lattice parameter is 5.484 Å (a = b = c), which is in agreement with the formation of a solid solution. Indeed, the value of the refined cell parameter is very close from the expected value (a = 5.480 Å) when a 50% solid solution is formed between $\alpha$-NaYF$_4$ (a = 5.507 Å) and $\alpha$-NaYbF$_4$ (a = 5.452 Å). The line profile analysis was performed by applying the whole powder pattern modelling (WPPM) approach[25,26], yielding an average crystallite size of 8 nm with a standard deviation of 1.1 nm using a lognormal distribution function, which is in agreement with the size distribution extracted from STEM. The WPPM approach was also used to determine the micro-strain in the starting core NCs, which is 0.03%. Low-dose high-resolution HAADF-

STEM image confirms that core NCs are single crystalline (Fig. 1e). The integrated differential phase contrast (iDPC)-STEM image[27,28] (Fig. 1f) enables the visualization of both the light (F atoms) and heavy (RE atoms) atomic columns together. The corresponding HAADF-STEM images are shown in the Supplementary Fig. 1. These optically active core NCs were utilized as starting seeds for the subsequent growth of optically inactive homogeneous or heterogeneous shell domains.

### Growth of homogeneous and heterogeneous shell domains: structural characterization

$\alpha$-NaYF$_4$ ($Fm\bar{3}m$, a = b = c = 5.507 Å) and CaF$_2$ ($Fm\bar{3}m$, a = b = c = 5.462 Å) were selected as the homogeneous and heterogeneous shell domains, respectively. Both were grown on the previously described $\alpha$-NaYF$_4$:Yb:Er core NCs following the exact same experimental protocol except for the cation composition of the shell precursor. PXRD patterns confirm that the cubic crystal structure of the starting core NCs is preserved, both after the growth of the homogeneous and heterogeneous shell domains, with no evidence of secondary phases (Fig. 2a-b).

HAADF-STEM images after shell growth confirm the formation of isotropic particles with sizes of 22.9±3.0 nm and 14.8±2.8 nm for the homogeneous and heterogeneous domains, respectively (Fig. 2c-d). Additionally, low magnification HAADF-STEM images reveal the formation of highly faceted (hexagonal shape) particles in the case of the homogeneous shell domain (Fig. 2c), while rounded-corner cubes are formed for its heterogeneous counterpart (Fig. 2d). It is also worth noting that the heterogeneous NCs have a clear contrast difference between the expected core and shell domains while homogeneous ones have a quasi-uniform contrast. Due to the obvious contrast difference between the assumed core and shell regions on the HAADF-STEM image obtained for the heterogeneous NCs (Fig. 2d), it is possible to extract the size of the core region only. The corresponding size distribution histogram of the core region is given in the Supplementary Fig. 3 and is in very good agreement with the one obtained for the starting core NCs (Fig. 1c).

The line profile analyses – by applying the WPPM approach (lognormal distribution function) – yield an average crystallite size of 22 nm and 14 nm for the homogeneous and heterogeneous structures, respectively. The average crystallite size distributions obtained by PXRD are in good agreement with the size distributions obtained from HAADF-STEM images. Nevertheless, compared to the starting core NCs, the micro-strain (obtained by the WPPM approach) increases from 0.03% (starting core NCs) up to 0.04% and 0.07% for the homogeneous and heterogeneous shell domains, respectively. The respective iDPC-STEM images (Fig. 2e-f) show the arrangement of heavy and light atoms in the structure.

### Homogeneous *vs*. heterogeneous structures: the difference between alloying and phase segregation

To determine the spatial distribution of chemical elements, energy-dispersive X-ray spectroscopy (EDS) maps (Fig. 3) of different chemical elements were acquired. Contrary to the anion (F) network, which is the same for all core and shell domains, the spatial distribution of various cations can be utilized to determine whether the anticipated core-shell structures are likely to form. As expected, Y is distributed throughout the whole volume of the particle when growing a homogeneous shell domain (Fig. 3c), while it is more localized in the case of the heterogeneous shell domain (Fig. 3d). The dark contrast in the centre of each homogeneous particle is due to the void formation during exposure of the electron beam for the duration of the EDS scan (Supplementary Fig. 4). The spatial distribution of Yb is particularly interesting because it is the only cation (with a high concentration), which is supposed to be confined only in the core region regardless of whether homogeneous or heterogeneous shell domains are grown. Interestingly, the corresponding maps reveal a huge difference

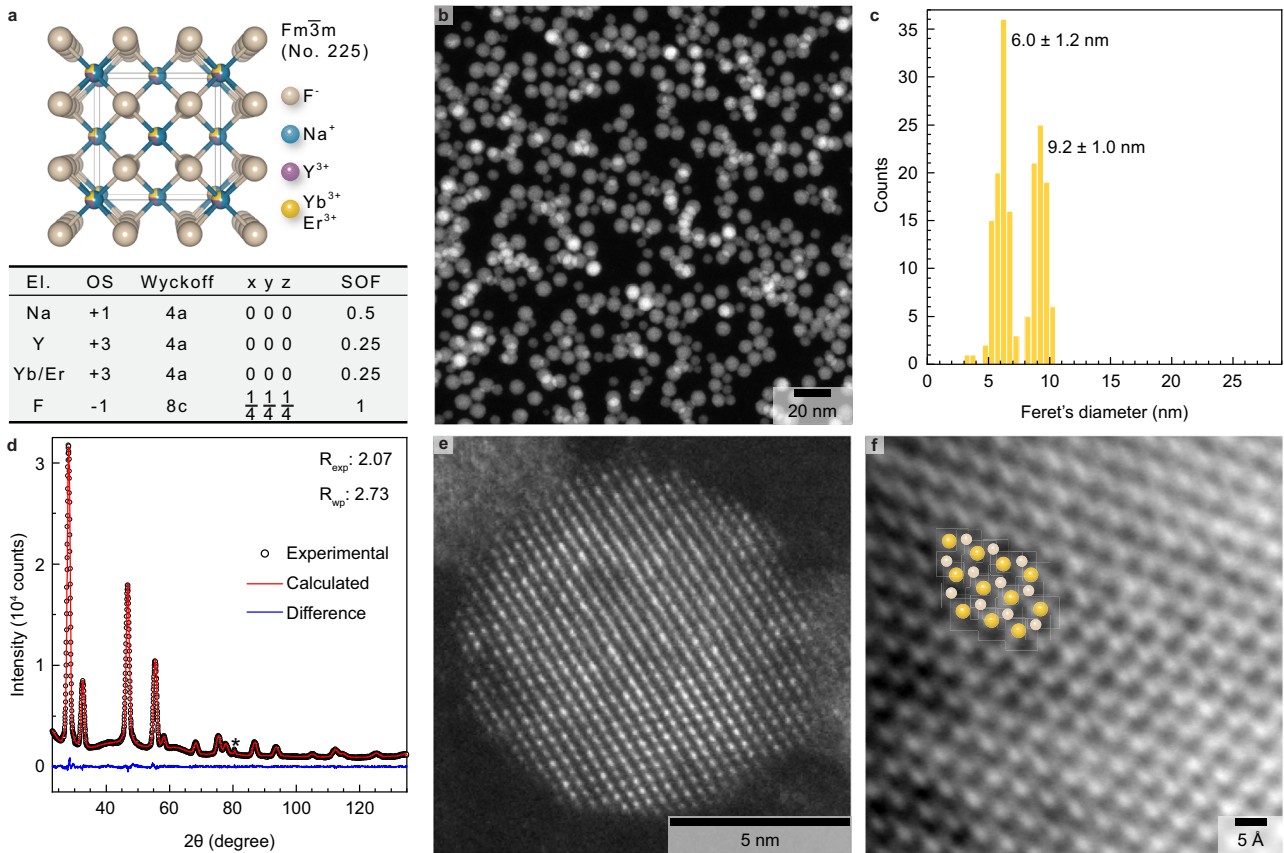

**Fig. 1 | Structural characterization of optically active α-NaYF₄:Yb:Er core nanocrystals (NCs). a** Cubic crystal structure (top part) of the starting α-NaYF₄:Yb:Er core NCs along the [001] direction (perspective view) together with important crystallographic data (bottom part) such as atomic positions (x y z) and site occupation frequency (SOF). **b** Representative low magnification high-angle annular dark-field scanning transmission electron microscopy (HAADF-STEM) image. At least ten HAADF-STEM images were acquired at various magnifications and spatial locations on the grid. The synthesis of core particles (without Ce³⁺ doping) was performed twice with similar results. **c** Corresponding size distribution histogram. **d** Experimental (black open symbols) and calculated (red solid line) X-ray powder diffraction patterns together with their difference curve (blue solid line). The calculated powder X-ray diffraction (PXRD) pattern is the result of a full Rietveld refinement. The values for the expected R factor ($R_{exp}$) and weighted profile R factor ($R_{wp}$) are given to assess the quality of the Rietveld refinement when combined with the visual inspection of the difference curve. Note that the peak at $2\theta \approx 80°$ (*symbol) does not belong to the cubic crystal structure. Nevertheless, it is properly described by the calculated PXRD pattern because the background was modelled by scaling the signal of the sample holder. The latter is a low background (911)-oriented silicon wafer onto which NCs are drop casted to obtain, after solvent evaporation, a thin film of NCs. The PXRD pattern of the empty sample holder is given in the Supplementary Fig. 2. **e** High-angle annular dark-field (HAADF-STEM) image of an individual α-NaYF₄:Yb:Er core NC. **f** Integrated differential phase contrast (iDPC)-STEM image of a region of an individual α-NaYF₄:Yb:Er core NC. Atomic columns that contain rare-earth and fluorine atoms are in yellow and beige, respectively.

between the homogeneous and heterogeneous shell domains. Indeed, in the former case, Yb is spatially diluted within the whole volume of the particle (Fig. 3e), while in the latter it is well confined and concentrated in a restricted central region of the particles (Fig. 3f). The corresponding overlay composites between the Y/Yb (Fig. 3g) or Yb/Ca (Fig. 3h) chemical maps clearly indicate that the anticipated core-shell structure is only probable (under the reported experimental conditions) in the case of the heterogeneous structure.

**Atomic scale organization of homogeneous and heterogeneous structures**

To map the atomic-scale organization of the NCs when growing either homogeneous or heterogeneous domains on the starting core NCs, local chemical analyses were performed on individual NCs. High-resolution HAADF-STEM images confirm that the NCs obtained after the growth of α-NaYF₄ (Fig. 4a) and CaF₂ (Fig. 4e) shell domains are single crystalline.

The high-resolution HAADF-STEM images also confirm the existence of an easily identifiable interface after the growth of CaF₂ contrary to the growth of NaYF₄. The quantitative analysis of HAADF-STEM

images, both for the homogeneous and heterogeneous NCs, is utilized to extract chemical information for each individual atomic columns within a single particle. The method relies on the accurate quantification of the intensity of different atomic columns in the corresponding high-resolution HAADF-STEM images. With the StatSTEM software[29], the projected atomic columns are modelled by a sum of Gaussian functions peaked at the individual atomic column positions. The volumes under those Gaussian peaks correspond to the scattering cross sections (SCS) that scale with the chemical composition and thickness of a projected atomic column, which is experimentally observed in the corresponding HAADF-STEM image (Fig. 4a, e)[30].

The distribution of the SCSs is then used to classify the atomic columns into groups depending on the averaged chemical composition. For the homogeneous structure, the analysis performed with a Gaussian mixture model indicates the presence of one single Gaussian component (Fig. 4b – orange solid line). This implies that the NC is alloyed and that heavy Yb atoms are observed over the whole volume of the NC. For the heterogeneous structure, the distribution of SCSs is described by two Gaussian components (Fig. 4f). The first component (Fig. 4f – green solid line) contains SCSs of the atomic columns

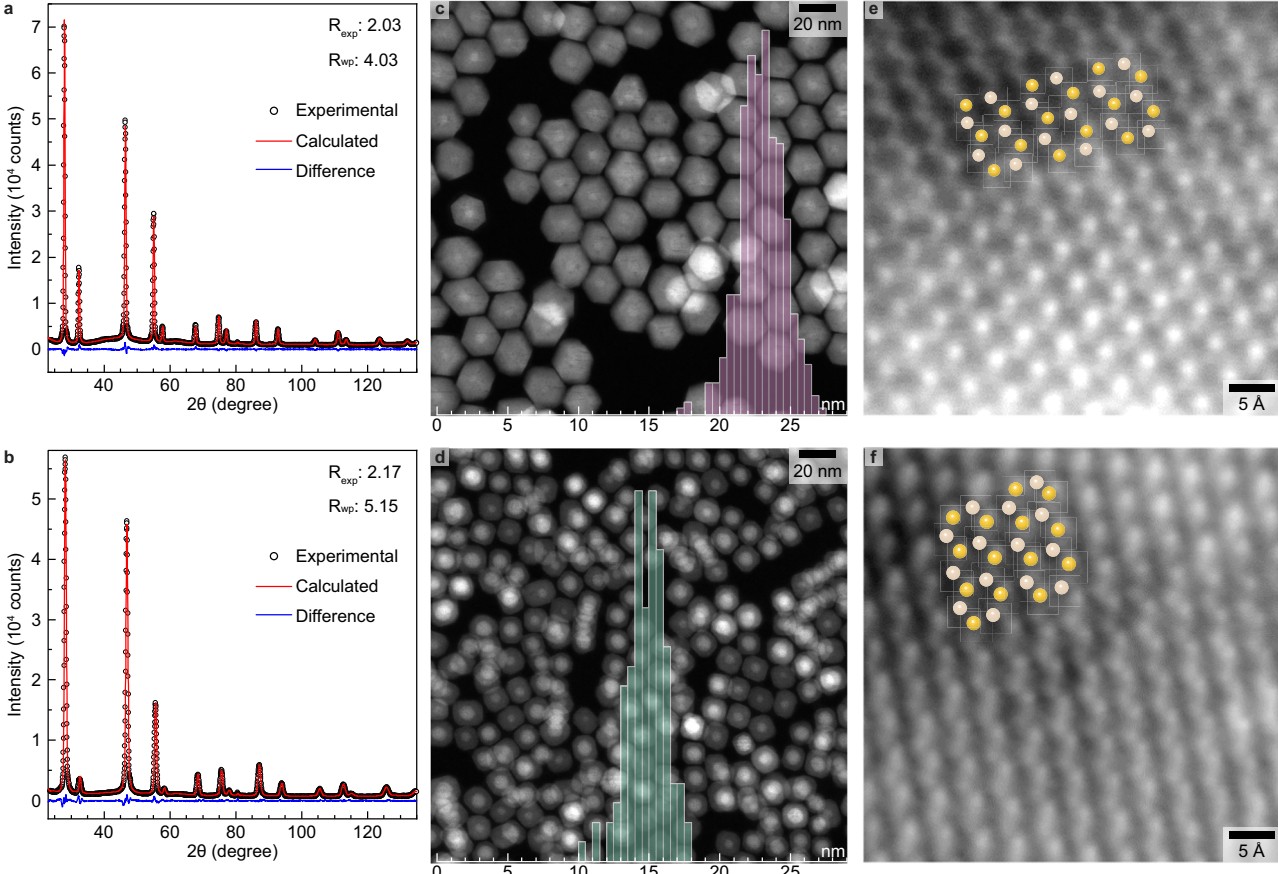

**Fig. 2 | Structural characterization of optically active α-NaYF₄:Yb:Er core nanocrystals (NCs) after growing homogeneous α-NaYF₄ (top row) and heterogeneous CaF₂ (bottom row) shell domains. a,b** Experimental (black open symbols) and calculated (red solid lines) powder X-ray diffraction (PXRD) patterns together with their difference curve (blue solid lines). The calculated PXRD patterns are the result of full Rietveld (**a**) and Pawley (**b**) refinements. The values for the expected R factor (R_exp) and weighted profile R factor (R_wp) are given to assess the quality of the refinements when combined with the visual inspection of the difference curve. Note that the peak at 2θ ≈ 80° (*symbol) does not belong to the cubic crystal structure. Nevertheless, it is properly described by the calculated PXRD pattern because the background was modelled by scaling the signal of the sample holder. The latter is a low background (911)-oriented silicon wafer onto which NCs are drop casted to obtain, after solvent evaporation, a thin film of NCs. The PXRD pattern of the empty sample holder is given in the Supplementary Fig. 2. **c**, **d** Representative low magnification high-angle annular dark-field scanning transmission electron microscopy (HAADF-STEM) images overlaid with their corresponding size distribution histograms. At least ten HAADF-STEM images were acquired at various magnifications and spatial locations on the grid for both the homogeneous (**c**) and heterogeneous (**d**) particles. The growth of a homogeneous shell domain on core particles (without Ce³⁺ doping) was performed twice with similar results. The growth of a heterogeneous shell domain on core particles (without Ce³⁺ doping) was performed four times with similar results. **e, f** Integrated differential phase contrast (iDPC)-STEM images of a region of individual homogeneous (**e**) and heterogeneous (**f**) NCs. Atomic columns that contain rare-earth and fluorine atoms are in yellow and beige, respectively. The exact same α-NaYF₄:Yb:Er core NCs (reported in Fig. 1) and experimental conditions were utilized for the growth of homogeneous and heterogeneous shell domains.

containing Ca atoms in the atomic columns. The second component (Fig. 4f – orange solid line) corresponds to SCSs of atomic columns containing at least one Yb atom. Each SCS can be assigned to the component having the highest probability for that particular SCS, thus resulting in a colorized map. The atomic columns corresponding to the shell and core are shown as green or orange dots overlaying their corresponding HAADF-STEM images (Fig. 4c, g).

The results obtained by the quantitative analysis of high-resolution HAADF-STEM images were further confirmed by spatially resolved EDS line-scan analysis. This technique, although limited in terms of spatial resolution to the sub-nanometer level, is complementary to the quantitative analysis of high-resolution HAADF-STEM images because it gives the chemical composition at each of the measured points, instead of classifying each atomic column based on its composition. Thus, combining the two methods yields an accurate representation of the atomic scale organization within an individual NC. It is clear that the Y (Fig. 4d – purple open circles) and Yb/Er chemical profiles (Fig. 4d – orange open circles), after growing the homogeneous shell domain, are not compatible with the formation of

a core-shell structure. Although a very thin (≤1 nm) outer layer of pure Y was detected, the rest of the NC contains RE elements (Y/Yb/Er) with a chaotic and non-constant distribution, which refutes the existence of an intact core domain. The situation is completely different after growing the heterogeneous shell domain. Indeed, the chemical profiles clearly demonstrate that Ca (Fig. 4h – green open circles) and Y/Yb/Er (Fig. 4h – orange open circles) atoms are perfectly segregated. While Ca is confined in a comparatively thick (about 3 nm) outer region, RE elements are confined in an inner region whose diameter (about 7.5 nm) is in good agreement with the size of the starting core NCs. Equally important, the interface between the outer and inner regions is easily identified and relatively abrupt (≤1 nm), as expected for a true core-shell structure.

### Preventing cation intermixing dramatically boosts the optical performance

The SWIR emission spectra of α-NaYF₄:Yb:Er core NCs (Ce³⁺-undoped and Ce³⁺-doped) after growing homogeneous (NaYF₄) and heterogeneous (CaF₂) shell domains under 980 nm excitation exhibit typical

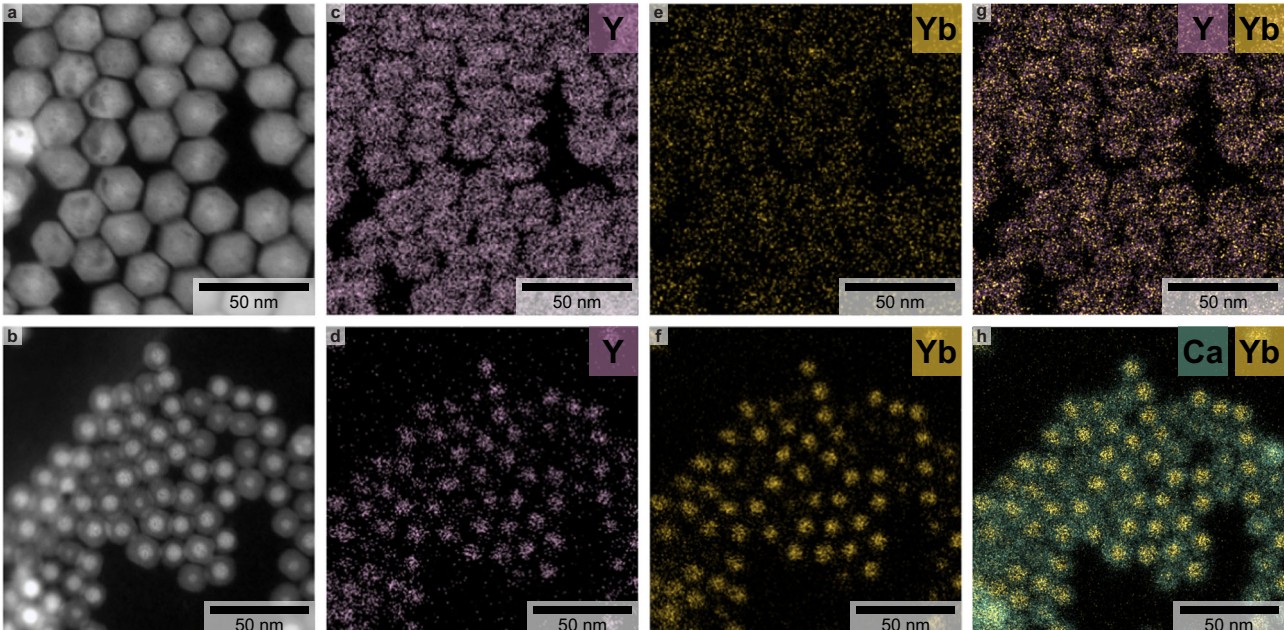

**Fig. 3 | Energy dispersive spectroscopy maps showing the yttrium and ytterbium distribution after growing homogeneous α-NaYF₄ (top row) and heterogeneous CaF₂ (bottom row) shell domains on optically active α-NaYF₄:Yb:Er core nanocrystals (NCs).** Representative low magnification high-angular annular dark-field scanning transmission electron microscopy (HAADF-STEM) images (**a, b**), yttrium chemical maps (**c, d**), ytterbium chemical maps (**e, f**), and overlay composites between the Yb/Y (**g**) or Yb/Ca (**h**) chemical maps.

Yb³⁺ ($^2F_{5/2} \rightarrow {}^2F_{7/2}$) and Er³⁺ ($^4I_{13/2} \rightarrow {}^4I_{15/2}$) bands, which are centred at 1015 nm and 1530 nm, respectively (Fig. 5a). The corresponding absolute PLQY in the power range 50-1000 mW/cm² (Fig. 5b) for the Er³⁺ SWIR emission (1400-1650 nm) is dramatically enhanced when growing a heterogeneous shell domain (PLQY = 30%) compared to the utilization of a homogeneous shell domain (PLQY = 7%). Even when considering the total SWIR emission range (1000-1650 nm) that includes both the emission from Yb³⁺ and Er³⁺, the absolute PLQY of the heterogeneous NCs (PLQY = 41%) is still more than double the efficiency of their homogeneous counterparts (PLQY = 17%). To boost the performance of the heterogeneous shell design, it was hypothesized that the absolute PLQY could be further improved by co-doping the starting core NCs with Ce³⁺. Indeed, it is well known that the Er³⁺/Ce³⁺ cross-relaxation processes can be used to enhance the SWIR emission of Er³⁺[9]. Therefore, an additional batch of optically active core NCs containing Yb/Er/Ce (42/5/3 at.%) was utilized to grow the homogeneous and heterogeneous shell domains (Supplementary Fig. 5) under the exact same conditions as for Ce-undoped core NCs. The emission spectra (Fig. 5a) are similar to the ones obtained with Ce-undoped core NCs but, as expected, the absolute PLQY is significantly enhanced relative to their Ce-undoped counterparts (Fig. 5b). Nevertheless, it is worth noting that, despite Ce doping, the efficiency of the NCs with a homogeneous shell domain is still lower compared to their Ce-undoped counterparts with a heterogeneous shell domain (24% vs. 30%). When the design is optimized (Ce-doped combined to heterogeneous shell) the NCs exhibit the highest reported PLQY (50%) for SWIR probes with a size of <15 nm, which was realized at a low excitation power density of 60 mW/cm². The latter is an order of magnitude lower than the skin exposure limit (730 mW/cm² at 980 nm)[24]. The trend is confirmed by fluorescence signal measurements (Fig. c) and imaging (Fig. d-g) under 980 nm excitation.

To display the biological contrast provided by the SWIR luminescence of the heterogeneous α-NaYF₄:Yb:Er:Ce-CaF₂ core-shell NCs, along with a relevant biomedical application enabled by it, a blood pool agent was formulated with α-NaYF₄:Yb:Er:Ce-CaF₂ micelles. In vivo macroscopic SWIR fluorescence imaging (1450 nm long pass) favoured the visualization of the fine mouse vasculature at high contrast when the α-NaYF₄:Yb:Er:Ce-CaF₂ micelles were injected intravenously (Fig. 5h-i and Supplementary Figs. 6 and 7).

## Discussion

To date, the vast majority (*ca.* 90%) of RE-based core-shell structures developed for SWIR downshifting rely on homogeneous ternary alkali metal fluorides (NaREF₄) both for the core and shell domains[14,15,31,32]. With the rapid democratization of the synthesis of such archetypes, a number of fundamentals have been neglected. This is the case for cation intermixing during shell growth, which is still widely overlooked despite its detrimental effect on the real atomic scale organization of expected structures and available energy migration pathways.

In this study, compelling experimental evidence revealed that the growth of homogeneous and heterogeneous shell domains on the same core NCs and under the exact same experimental conditions leads to a huge difference in the resulting atomic scale organization of the final composite. Indeed, when growing a homogeneous shell domain, the integrity of the starting core NCs is destroyed. The optically active centers, supposed to be confined in the core domain, are redistributed in the entire particles' volume leading to the formation of a disordered alloy instead of the hypothesized core-shell structure. On the contrary, such a phenomenon is not observed when growing a heterogeneous shell domain that enables the formation of a true core-shell structure with an abrupt and relatively thin (<1 nm) interface. The heterogeneous shell growth on the optically active core NCs enables to maintain the structural and chemical integrity (in terms of not only composition but also concentration) of both domains while efficiently preventing the formation of unwanted energy migration pathways to surface quenchers. The true chemical segregation observed for the heterogeneous structures instead of the uncontrolled alloying observed with the homogeneous ones enables to dramatically boost the PLQY. As indicated in Table 1, the heterogeneous RE-based core-shell NCs presented in this article are, under limited irradiance (only 60 mW/cm²), the most efficient SWIR emitters ever reported within the sub-15 nm size regime. It is worth noting that further gains

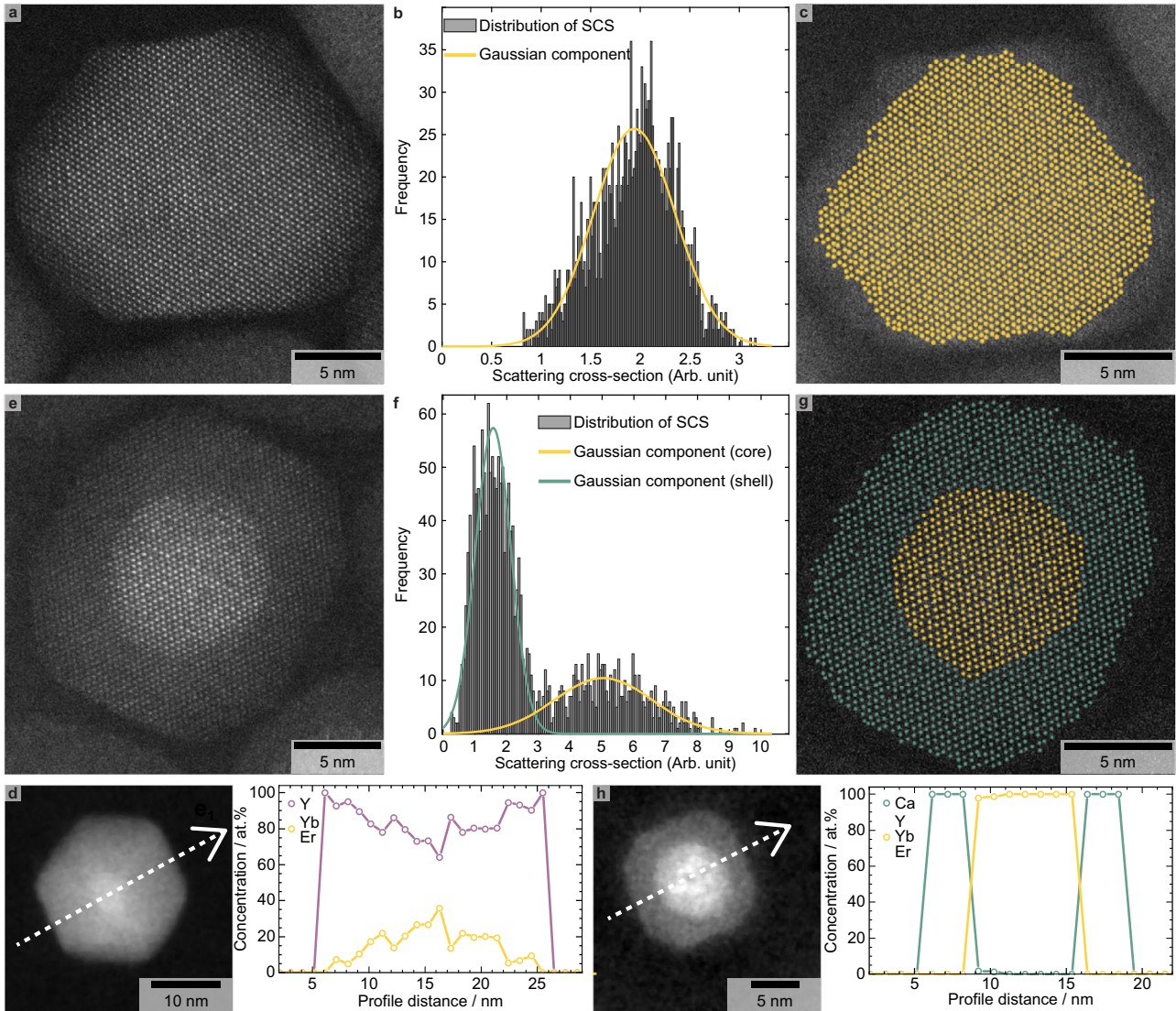

**Fig. 4 | Local chemical composition of optically active α-NaYF₄:Yb:Er core nanocrystals (NCs) after growing homogeneous α-NaYF₄ (a-d) and heterogeneous CaF₂ (e-h) shell domains. a/e** High resolution high-angle annular dark-field scanning transmission electron microscopy (HAADF-STEM) image of individual NCs. **b/f** Scattering cross section histograms with the classification into groups using a Gaussian mixture model with components. **c/g** Refined high-resolution HAADF-STEM images (using the StatSTEM software) overlaid by maps (colorized dots) reflecting the probability of finding at least one Yb atom (orange dots) or one Ca atom (green dots) for each atomic column within an individual homogeneous (**c**) or heterogeneous (**g**) NC. **d/h** HAADF-STEM images of individual particles together with their corresponding chemical concentration profiles. The latter were obtained by implementing the subshell approach (cf. Online Methods)[18,39], which was used to process the raw compositions obtained from quantified energy dispersive X-ray spectroscopy (EDS) spectra (white dashed arrows on the

corresponding HAADF-STEM images). Such a data treatment eliminates the superposition of the shell and core contributions to the overall EDS signal and yields the local chemical composition of each subshell and, hence, the real chemical composition for each measured data point. In the case of the deposition of the homogeneous shell domain, the yttrium and ytterbium/erbium concentration profiles are in purple and yellow, respectively (**d**). Similar concentration profiles have been measured on seven individual particles that were randomly selected. In the case of the deposition of the heterogeneous shell domain, the calcium and yttrium/ytterbium/erbium concentration profiles are in green and yellow, respectively (**h**). Similar concentration profiles have been measured on eight individual particles that were randomly selected. The exact same α-NaYF₄:Yb:Er core NCs (reported in Fig. 1) and experimental conditions were utilized for the growth of homogeneous and heterogeneous shell domains.

could yet be made (while maintaining the sub-15 nm size) via the optimization of doping concentrations and shell thickness. This is beyond the scope of the present work, given that time-consuming nature of both synthesis and atomic scale characterization. Indeed, not only series of core NCs must be synthesized by systematically varying the concentration of one optically active element (Yb³⁺, Er³⁺ or Ce³⁺), while maintaining the other two constant, but also every series should be coated by a protective shell with different thicknesses. Such a task might soon be solved with the emergence of automated synthesis laboratories coupled to artificial intelligence techniques (e.g. machine learning algorithms) in the field of inorganic NCs[33].

The results presented in this study can also explain past results obtained with heterogeneous core-shell structures. In particular, the pioneering works performed by Wang and co-workers on upconversion enhancement in heterogeneous α-NaYF₄:Yb:Er-CaF₂ core-shell NCs compared to their homogeneous α-NaYF₄:Yb:Er-NaYF₄ counterparts[34]. The authors hypothesized that trivalent RE elements (Yb, Er) could have diffused into the homogeneous NaYF₄ shell, whereas the process could have been suppressed in the heterogeneous CaF₂ shell due to the different oxidation states. Similar observations have been reported for near-IR to ultraviolet upconversion when comparing heterogeneous α-NaYbF₄:Tm-CaF₂ and homogeneous

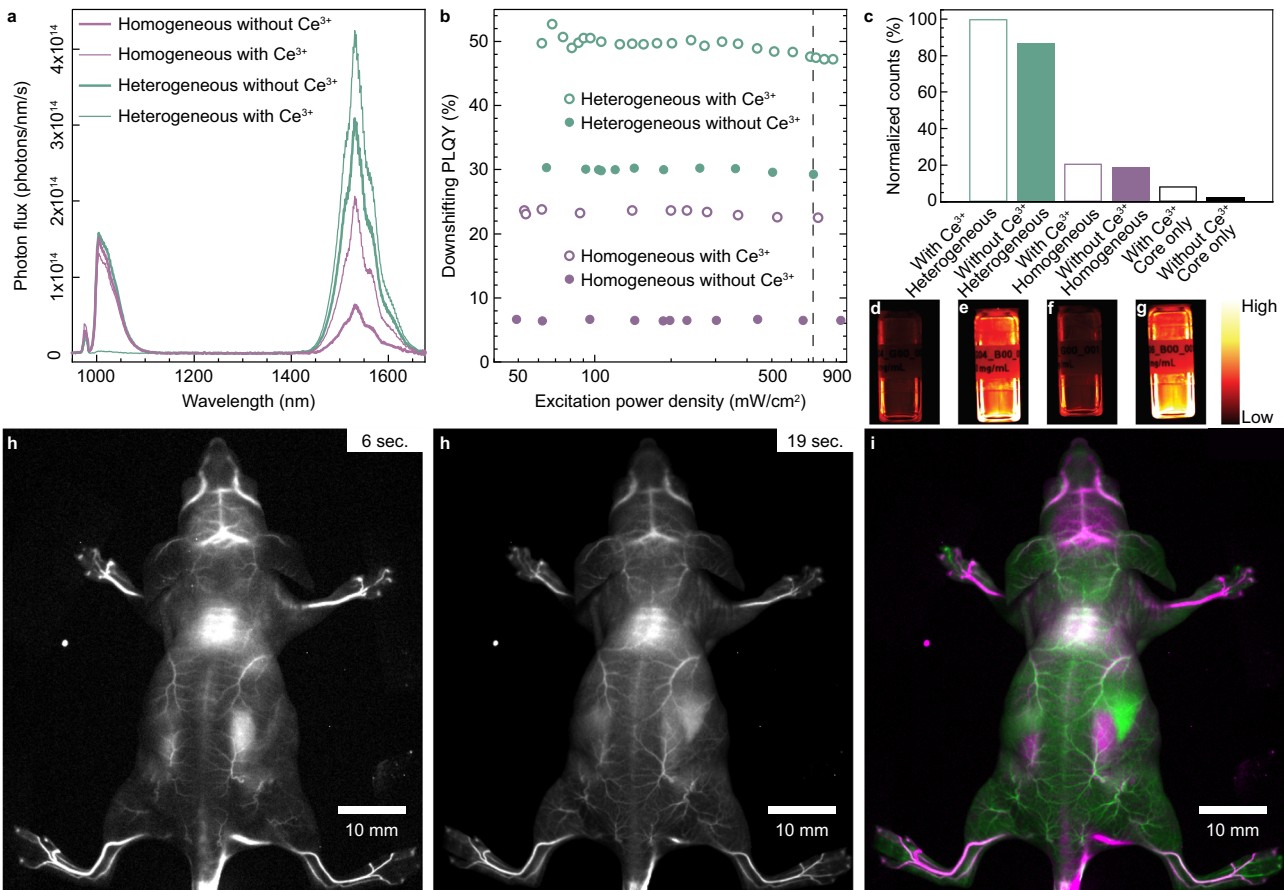

**Fig. 5 | Optical performance after growing homogeneous α-NaYF₄ and heterogeneous CaF₂ shell domains on optically active α-NaYF₄:Yb:Er and α-NaYF₄:Yb:Er:Ce core nanocrystals (NCs). a** Absorption-normalized downshifting (λ_ex. = 980 nm) emission spectra (@200 mW/cm²) for α-NaYF₄:Yb:Er (thick solid lines) and α-NaYF₄:Yb:Er:Ce (thin solid lines) core NCs after growing homogeneous (purple solid lines) and heterogeneous (green solid lines) shell domains . **b** Absolute downshifting photoluminescence quantum yield (PLQY) in the power range 50–1000 mW/cm² with λ_ex. = 980 nm and 1400 <λ_em. < 1650 nm for α-NaYF₄:Yb:Er (filled symbols) and α-NaYF₄:Yb:Er:Ce (open symbols) core NCs after growing homogeneous (purple symbols) and heterogeneous (green symbols) shell domains. **c**, Fluorescence signal measurements (normalized counts) under 980 nm excitation (37 mW/cm²) for α-NaYF₄:Yb:Er (black solid bar) and α-NaYF₄:Yb:Er:Ce (black open bar) core NCs with homogeneous (purple bars) and heterogeneous

(green bars) shell domains. The bottom panel shows images of the emission (1300 nm long pass filter) of homogeneous (**d, f**) and heterogeneous (**e, g**) core-shell NCs without (**d, e**) and with (**f, g**) Ce³⁺ doping under 980 nm excitation (37 mW/cm²). All fluorescence signal measurements were performed on colloidal toluene dispersions with a concentration of 10 mg/mL. **h** In vivo SWIR fluorescence imaging of α-NaYF₄:Yb:Er:Ce-CaF₂ micelles injected in a mouse tail vein. Diffuse illumination (968 nm) was used for excitation (mean power density 100 mW/cm²), and InGaAs detection was performed with a 1450 nm long pass filter. Single frames of different elapsed injection time points at the same scale. **i** Multicolor angiography image with color separation using principal component analysis based on vascular kinetics of the injected NCs. The magenta and green colors represent earlier and later arrival times, respectively. Liver and kidneys are visible in the middle right part of the image in bright green and magenta colors, respectively.

β-NaYF₄:Yb:Tm-NaYF₄ core-shell NCs. The improved ultraviolet emission (in water) of the former compared to the latter was nonetheless attributed to the fact that the CaF₂ shell was a less disordered and better crystalline phase than β-NaYF₄[35]. More recently, Tan and coworkers investigated near-IR downshifting (800→1000 nm) in heterogeneous α-NaYF₄:Yb:Nd-CaF₂ core-shell NCs[36].

Although the authors did not compare the near-IR performance with homogeneous core-shell NCs, they interestingly observed that the near-IR emission at 1000 nm nearly completely vanished when the sensitizer (*i.e.* Nd³⁺) and activator (*i.e.* Yb³⁺) were separated into the shell and core domains, respectively. Assuming that the chemical integrity of the core and shell domains is also retained in the heterogeneous NCs reported by Tan and coworkers[36], the 10-times weaker emission observed (spatial confinement instead of co-doping) at 1000 nm is a sign that the energy transfer rate between optically active centers can be modified depending on their relative spatial distribution. To validate such an assumption, additional ion-ion energy transfer calculations would need to be performed.

Our investigations highlights the major benefit of heterostructures over homogeneous ones for preventing cation intermixing. This enables to synthesize true core-shell structures (instead of disordered alloys) to maximize their SWIR performance in a size regime that is compatible with in vivo applications[37]. The fact that cation intermixing between different domains can be avoided in heterogeneous NCs dramatically affects the true local concentration of optically-active centers, which is not trivial to measure accurately. The direct consequence is that optimum concentrations of the sensitizer, activator, and optically-inactive elements, which were mainly determined with homogeneous NCs, might no longer be valid for their heterogeneous counterparts. Similarly, the optimum thickness of the inert outer shell to prevent luminescence surface quenching should be reassessed. This might explain why recently published heterostructures for SWIR imaging did not reach similar performances to the one reported in this manuscript despite the utilization of heterostructures[38]. The synthetic strategy that was pioneered a decade ago by Yan and co-workers[34], should be seriously considered as a promising route to enhance the photoluminescence properties of RE elements in inorganic NCs. Although CaF₂ offer

**Table 1 | Comparison of the optical performances of various rare-earth based nanocrystals for SWIR emission (within each sub-category, results are ordered chronologically)**

| Core domain[a] | Shell domain[a] | Overall size [b] (nm) | Excitation (nm) | Emission (nm) | PLQY[c] (%) | Power density (mW/cm²) | Reference |
|---|---|---|---|---|---|---|---|
| *Unprotected (i.e. no shell) core structures* | | | | | | | |
| β-NaCeF$_4$:Yb:Er | NA | 200 | 980 | 1525 | 32 | | 46 |
| β-NaCeF$_4$:Yb:Er | NA | 6<br>25 | 980<br>980 | 1525<br>1525 | 6<br>18 | 100000 | |
| β-NaNdF$_4$:Mn | NA | 5 | 785 | 1050 | 10[d] | N/S | 47 |
| CaF$_2$:Nd | NA | 10-15 | 808 | 1020-1100 | 9[e] | 483 | 48 |
| α-CaS:Ce:Er | NA | 25 | 450 | 1540 | 9 | N/S | 49 |
| α-CaS:Ce:Nd | NA | 25 | 450 | 1070 | 7 | | |
| β-NaLuF$_4$:Gd:Yb:Er:Ce | NA | 100 × 20 | 980 | 1450-1650 | 26<br>**4** | N/S | 50 |
| *Expected core-shell structures with homogeneous core and shell domains* | | | | | | | |
| β-NaYF$_4$:Yb:Er | β-NaLuF$_4$ | 29 | 980 | 1450-1650 | 18 | 1700 | 51 |
| β-NaYF$_4$:Yb:Er | β-NaGdF$_4$ (inner)<br>β-NaGdF$_4$ (outer) | 52 | 800 | 1000 (main)<br>1550 (minor) | (13) | 200 | 52 |
| β-NaYbF$_4$:Er:Ce | β-NaYF$_4$ | 18 | 980 | 1300 – 1900 | (23)<br>**(3)** | 150 | 9 |
| β-NaErF$_4$ | β-NaLuF$_4$ | 40 | 808 | 1525 | 11 | N/S | 53 |
| α-NaYbF$_4$:Er:Ce:Zn | α-NaYF$_4$ | 14 | 970 | 1525 | **5** | 100 | 11 |
| NaErF$_4$ | NaYbF$_4$ (inner)<br>NaYF$_4$ (outer) | 30 | 980 | 1525 | 19 | N/S | 54 |
| β-NaYF$_4$:Nd | β-NaYF$_4$ | 66 | 808 | 1064<br>1345 | (8)<br>(5) | N/S | 55 |
| β-NaErF$_4$ | β-NaYF$_4$ | 13 | 808 | 1525 | 10 | 660 | 56 |
| β-NaErF$_4$ | β-NaLuF$_4$ | 30 × 25 | 800 | 1550 | 14 | 880 | 57 |
| LiYbF$_4$:Er:Ce | LiYF$_4$: | 17 | 980 | 1450-1650 | 36 | 12500 | 58 |
| α-NaYF$_4$:Yb:Er:Ce | α-NaYF$_4$ | 45 | 980 | 1450-1650 | 24 | 60 | This work |
| α-NaYF$_4$:Yb:Er | α-NaYF$_4$ | 22 | 980 | 1450-1650 | 7 | | This work |
| *Expected core-shell structures with heterogeneous core and shell domains* | | | | | | | |
| α-NaYF$_4$:Yb:Nd | CaF$_2$ | 10 | 800 | 950-1100 | 11 | 1000 | 36 |
| α-NaYbF$_4$:Er:Ce:Zn | CaF$_2$ | 19<br>28<br>53<br>81 | 980 | 1450-1650 | 22<br>29<br>40<br>49 | 20000 | 38 |
| α-NaYF$_4$:Yb:Er:Ce | CaF$_2$ | 15 | 980 | 1450-1650 | 50 | 60 | This work |
| α-NaYF$_4$:Yb:Er | CaF$_2$ | 15 | 980 | 1450-1650 | 30 | | This work |

[a]Cubic (α) and hexagonal (β) phases when the information is available in the corresponding original article.
[b]One or two values for isotropic and anisotropic shapes, respectively.
[c]Bold and normal text indicate that quantum yield measurements have been performed in aqueous solution or organic solvents, respectively. Quantum yield values determined by the relative method are in brackets (comparison with a reference dye) while those determined by the absolute method (integrating sphere) are normal text.
[d]No information is given on the original article on how the quantum yield was measured.
[e]Quantum yield measured for the solid-state powder. Each sub-section of the table is chronologically organized from the oldest (top) to the newest (bottom). N/S stands for Not Specified.

interesting possibilities, it represents only one combination that was tested amongst a wide range of potential combinations. Those hetero-structures, which still represent less than 10% of the reported RE-based core-shell NCs, constitute an important paradigm shift for the emergence of a new generation of highly efficient SWIR emitters with well-controlled atomic scale organization. Although the optimization of the concentration of RE elements and inert shell thickness might be of interest, the authors believe that the room for improvement will very likely be marginal without exploring truly disruptive core-shell, segmented or dimer heterostructures. The latter offer a unique possibility to modify the cation and/or anion networks between the different nano-sized subdomains. This will lead to the ability to combine materials into functional RE-based architectures, which currently have no existing counterparts. Such a synergetic coupling of properties might be the key for significant improvement in performance in the field of RE photo-luminescence of nanomaterials.

## Methods

### Materials

All animal ethics and protocols were approved by and are in agreement with regulations of the government of Upper Bavaria. Octadecene (ODE, technical 90%), oleic acid (OA, technical grade, 90%), oleylamine (OAm, ≥98% primary amine), sodium trifluoroacetate (NaOOCCF$_3$, 98%), cerium(III) nitrate (Ce(NO$_3$)$_3$.6H$_2$O), trifluoroacetic acid (99%), absolute ethanol (EtOH), and toluene (C$_6$H$_5$CH$_3$) were purchased from Sigma Aldrich and were used without further purification. Calcium carbonate (CaCO$_3$, 99.99%) and rare-earth oxides (Y$_2$O$_3$, Yb$_2$O$_3$, Er$_2$O$_3$, min. purity 99.99%) were purchased from Alfa Aesar. 1,2-dioleoyl-sn-glycero-3-phosphoethanolamine-N-[methoxy(polyethylene glycol) −2000] (ammonium salt) (18:1 PEG2000 PE) was purchased from Avanti Polar Lipids. Note that ODE, OA, OAm, NaOOCCF$_3$, and Ce(NO$_3$)$_3$.6H$_2$O are stored under inert conditions in a glovebox under dry nitrogen (O$_2$ and H$_2$O levels <1 ppm) when received. All given

quantities for trifluoroacetate precursors were corrected based on their corresponding thermogravimetric analyses.

## Synthesis of rare-earth trifluoroacetates RE(OOCCF₃)₃ (RE = Y, Yb, Er)

RE(OOCCF₃)₃ were prepared by adding 20 mmol of rare-earth oxide (Y₂O₃, or Yb₂O₃, or Er₂O₃) to 14 mL of trifluoroacetic acid and 36 mL of deionized water in a 100 mL round bottom flask equipped with a reflux condenser. The stirred suspension was heated up to 95 °C under air until the solid oxide was completely dissolved and a clear colorless solution (Y, Yb) or pinkish solution (Er) was obtained. After cooling, the solution was filtered and the solvent evaporated with a rotary evaporator at 70 °C for 90 min. The obtained wet powder was dried under vacuum at 110 °C for 24 h. The dry powder was stored under inert conditions in a glovebox under dry nitrogen. Note that the obtained RE(OOCCF₃)₃ precursors were slightly hydrated with a water quantity <1 wt.% as determined by thermogravimetric analysis.

## Synthesis of calcium trifluoroacetate Ca(OOCCF₃)₂

Ca(OOCCF₃)₂ was prepared by adding 72 mmol of CaCO₃ together with 16 mL of trifluoroacetic acid and 33 mL of deionized water in a 100 mL round bottom flask equipped with a reflux condenser. The stirred suspension was heated up to 95 °C under air until the solid carbonate was completely dissolved and a clear colorless solution was obtained. After cooling, the solution was filtered and the solvent evaporated with a rotary evaporator at 70 °C for 180 min. The obtained wet powder was dried under vacuum at 110 °C for 24 h. The dry powder was stored under inert conditions in a glovebox under dry nitrogen. Note that the obtained Ca(OOCCF₃)₂ precursor was slightly hydrated with a water quantity of 0.8 wt.% as determined by thermogravimetric analysis.

## Synthesis of α-NaYF₄:Yb:Er core nanocrystals

In a glovebox under dry nitrogen, 4 mmol of NaOOCCF₃, 1.99 mmol of Y(OOCCF₃)₃, 1.81 mmol of Yb(OOCCF₃)₃, and 0.21 mmol of Er(OOCCF₃)₃, are introduced in a 250 mL three-neck round bottom Schlenk flask together with ODE (92.5 mmol), OA (39.8 mmol) and OAm (40.0 mmol). The resulting mixture is transferred to a Schlenk line and heated up to 110 °C (under Ar flow) to dissolve the trifluoroacetate precursors. The temperature is maintained for 30 min during which the turbid solution turns to an optically clear slightly yellowish solution. Then, the temperature is decreased to 100 °C, and the optically clear solution is purified under vacuum. The vacuum purification consists in five Ar ↔ vacuum (ca. 5.0.10⁻² mbar) cycles followed by a dynamic vacuum step (down to 5.10⁻⁴ mbar) for 10 min. Finally, the solution is heated up under Ar to 300 °C for 30 min (Supplementary Fig. 8). Then, the heating mantle is removed, and the flask is left to cool naturally to room temperature. After cooling, the NCs are extracted, purified, and finally dispersed in 10 mL toluene. The resulting clear colorless solution is stored in a tightly closed glass vial and utilized as a stock solution for the subsequent growth of homogeneous and heterogeneous core-shell NCs.

## Synthesis of α-NaYF₄:Yb:Er:Ce core nanocrystals

The synthesis is exactly the same as described for α-NaYF₄:Yb:Er core NCs except that 3 mol.% of Ce(NO₃)₃.6H₂O is also added as Ce³⁺ precursor. To keep constant the total quantity of RE elements, the quantity of Yb(OOCCF₃)₃ is reduced to 42 mol.% (instead of 45 mol.%).

## Synthesis of α-NaYF₄:Yb:Er@NaYF₄ homogeneous core-shell nanocrystals

In a glovebox under dry nitrogen, 2 mmol of NaOOCCF₃ and 2 mmol of Y(OOCCF₃)₃ are introduced in a 50 mL three-neck round bottom Schlenk flask together with ODE (22 mmol) and OA (22 mmol). The resulting mixture is transferred to a Schlenk line and 833 µL of the

stock solution of α-NaYF₄:Yb:Er core NCs are added. The resulting mixture is heated up to 120 °C (under Ar). The temperature is maintained for 30 min and then toluene is removed under reduced pressure by applying five Ar ↔ vacuum (5.0.10⁻² mbar) cycles. Finally, the solution is heated up under Ar to 300 °C for 45 min (Supplementary Fig. 8). Then, the heating mantle is removed, and the flask is left to cool naturally to room temperature. After cooling, the NCs are extracted, purified, and finally dispersed in 2 mL toluene.

## Synthesis of α-NaYF₄:Yb:Er@CaF₂ heterogeneous core-shell nanocrystals

The synthesis is exactly the same as described for α-NaYF₄:Yb:Er@NaYF₄ homogeneous core-shell NCs except that NaOOCCF₃ and Y(OOCCF₃)₃ are replaced by 2 mmol of Ca(OOCCF₃)₂.

## Synthesis of α-NaYF₄:Yb:Er:Ce@NaYF₄ homogeneous core-shell nanocrystals

The synthesis is exactly the same as described for α-NaYF₄:Yb:Er@NaYF₄ homogeneous core-shell NCs except that α-NaYF₄:Yb:Er core NCs are replaced by α-NaYF₄:Yb:Er:Ce core NCs.

## Synthesis of α-NaYF₄:Yb:Er:Ce@CaF₂ heterogeneous core-shell nanocrystals

The synthesis is exactly the same as described for α-NaYF₄:Yb:Er@NaYF₄ homogeneous core-shell NCs except that NaOOCCF₃ and Y(OOCCF₃)₃ are replaced by 2 mmol of Ca(OOCCF₃)₂. α-NaYF₄:Yb:Er core NCs are also replaced by α-NaYF₄:Yb:Er:Ce core NCs.

## Powder X-ray diffraction (PXRD)

PXRD was performed at room temperature in Bragg-Brentano geometry using a Bruker D8 Discover powder diffractometer with a copper anticathode, a quartz monochromator (Cu Kα1 radiation, λ=1.540562 Å), and a 1-dimensional detector (LynxEye XE-T). PXRD patterns were recorded in the 2θ range 10°–135° for 9 h. Line profile analyses and Rietveld refinements were performed by using TOPAS software (version 7). PXRD samples were prepared by drop-casting colloidal suspensions of the NCs precipitated in acetone onto low background (911)-oriented silicon substrates.

## EDX spectroscopy: line-scan analysis

The experiments were carried out with a FEI Osiris ChemiSTEM microscope (operated at 200 keV electron energy), which is equipped with a Super-X EDX system (comprising four silicon drift detectors - Bruker XFlash) for EDX spectroscopy. EDX spectra are quantified with the FEI software package "TEM imaging and analysis" (TIA) version 4.7 SP3. Using TIA, element concentrations were calculated on the basis of a refined Kramers' law model, which includes corrections for detector absorption and background subtraction. For this purpose, standard-less quantification (theoretical sensitivity factors) without thickness correction was applied. The quantification of F-, Na-, Ca-, Y-, Yb-, and Er-content from their EDX spectra (line scans) was performed by evaluating the intensities of the F-Kα line, Na-K series, Ca-K series, Y-L, Yb-L, and Er-L series. We noted that X-ray lines of Cu (K- and L-series) from the grid, Yb-M and Er-M series from the NCs, as well as the C-Kα line from the amorphous carbon substrate were always present in the EDX spectra. Grids were prepared at room temperature in air by drop casting 10 µL of a diluted suspension of NCs in toluene onto an ultra-thin amorphous carbon film (3 nm) on holey carbon support film mounted on 400 µm mesh Cu grid (Ted Pella Inc.).

Concentration profiles of different elements within a single nanocrystal were obtained from EDX spectra acquired along a line that passes through the center of the corresponding NC. EDX line profiles were recorded by applying a drift-correction routine via cross correlation of several images, which yields a local precision better than 1 nm.

The drift-corrected EDX line profiles were acquired with a probe diameter of 0.4 nm and a distance of 1 nm between two measuring points along the line.

Regarding the quantification of the EDX spectra of core-shell NCs, one has to keep in mind that the obtained compositions are averaged along the electron-beam direction. In other words, the whole volume along the electron beam trajectory contributes to the detected X-ray signal. This means that local EDX spectra acquired at different positions along the line scan lead to an average chemical composition (Supplementary Figs. 9a and 9b). To determine the true chemical composition of different regions of core-shell NCs, raw EDX data were treated by the subshell approach that was described in details elsewhere[18,39]. The subshell approach, which was successfully applied to investigate the chemical composition of quasi-spherical core-shell NCs for a variety of materials[18], was also implemented in this study to determine the local chemical composition of the homogeneous α-NaYF$_4$:Yb:Er–NaYF$_4$ core-shell NCs.

It is worth noting that the underlying mathematical model of the subshell approach[18,39] that was implemented for homogeneous α-NaYF$_4$:Yb:Er–NaYF$_4$ core-shell NCs cannot be utilized to determine the local chemical composition of α-NaYF$_4$:Yb:Er–CaF$_2$ NCs. Indeed, HAADF-STEM images clearly indicate that heterogeneous NCs have a significantly different morphology compared to their homogeneous counterparts and must be described as a quasi-cubic shell surrounding a quasi-spherical core. Therefore, to take into account the different geometrical shapes, the underlying mathematical model of the subshell approach was modified as described below and in the Supplementary Figs. 9c and 9d).

Similarly to the subshell approach utilized for quasi-spherical NCs, EDX spectra are acquired along a line that passes through the center of the heterogeneous NC. The step size $\Delta X$ between two successive measuring points is 1 nm. The first EDX spectrum is acquired at the spatial position $X_1 = \frac{\Delta X}{2}$ from the outer edge of the NC. The second EDX spectrum is measured at the spatial position $X_2 = X_1 + \Delta X$ and the $X_n$-th EDX spectrum is measured at the position $X_n = X_{n-1} + \Delta X$. Therefore, the global chemical composition is measured for each chemical element i (i = F, Na, Ca, Y, Yb, Er) for every $X_n^{th}$ position with a spatial resolution $\Delta X = 1 nm$. In the case of the heterogeneous α-NaYF$_4$:Yb:Er-CaF$_2$ core-shell NCs, raw EDX spectra for which only Ca and F are detected indicate the formation of pure CaF$_2$. The thickness of the pure CaF$_2$ domain is true until the position $X_k (1 \le k \le n)$. If $C_{X_j}^{Ca}$ and $C_{X_j}^{F}$ represent the concentration of Ca and F, which are determined by the quantification of the corresponding EDX spectra measured at every $X_j (1 \le j \le k)$ positions, the following elemental concentration relationships apply for all $X_j$ positions:

$$C_{X_1}^{Ca} = C_{X_2}^{Ca} = \cdots = C_{X_k}^{Ca} \tag{1}$$

$$C_{X_1}^{F} = C_{X_2}^{F} = \cdots = C_{X_k}^{F} \tag{2}$$

$$C_{X_j}^{Ca} + C_{X_j}^{F} = 1 \, with \, (1 \le j \le k) \tag{3}$$

$$C_{X_j}^{Na} + C_{X_j}^{Y} + C_{X_j}^{Yb} + C_{X_j}^{Er} = 0 \, with \, (1 \le j \le k) \tag{4}$$

For spatial positions $X_j > X_k (k + 1 \le j \le n)$, and for which all chemical elements i are detected, the formation of a spherical core region is assumed, which is in agreement with typical HAADF STEM images of NCs. The spatial extension of the core region is defined by the limits $X_{Spher.}^{Start} = X_k + \frac{\Delta X}{2}$ and $X_{Spher.}^{End} = a - (X_k + \frac{\Delta X}{2})$, where $a$ represents the length of the size edge of the cube (Supplementary Fig. 9a). Therefore, the core region within these limits must be divided into concentric

spherical subshells ($SS_1$ to $SS_{n-k}$) with a thickness $\Delta X = 1 nm$. The radius of the spherical region between $X_{Spher.}^{Start}$ and $X_{Spher.}^{End}$ is defined by $R = \frac{a}{2} - X_{Spher.}^{Start}$ (Supplementary Fig. 9c). Considering that $C_{X_k}^{i}$ is the concentration of the chemical element i obtained from the quantification of the corresponding EDX spectrum (acquired at position $X_k$) and assuming that the average chemical composition is obtained along the electron beam propagation, the concentration $C_{X_{k+1}}^{i}$ measured at the next position $X_{k+1}$ is the weighted concentration of element i measured at the position $X_k (\omega_{0;1}, C_{X_k}^{i})$ and its concentration within the first spherical subshell $(\omega_{1;1}, C_{SS_1}^{i})$. The weight $\omega_{0;1}$ is proportional to the corresponding thickness $d_{0;1}$ measured from the outer surface of the NC to the surface of the first subshell $SS_1$ along the electron beam propagation $X_{k+1}$. Similarly, the weight $\omega_{1;1}$ is proportional to the thickness $d_{1;1}$, which is defined as the thickness that is probed by the electron beam through the first subshell $SS_1$ at position $X_{k+1}$. Consequently, the concentration of the element i in the first subshell $C_{SS_1}^{i}$ can be calculated by using the following equations:

$$C_{SS_1}^{i} = \frac{C_{X_{k+1}}^{i} - \omega_{0;1} \bullet C_{X_k}^{i}}{\omega_{1;1}} = \frac{C_{X_{k+1}}^{i} - \frac{2d_{0;1}}{2d_{0;1} + 2d_{1;1}} \bullet C_{X_k}^{i}}{\frac{2d_{1;1}}{2d_{0;1} + 2d_{1;1}}} \tag{5}$$

$$d_{0;1} = \frac{a}{2} - d_{1;1} \tag{6}$$

$$d_{1;1} = \sqrt{R^2 - \left(R - \frac{1}{2} \bullet \Delta X\right)^2} \tag{7}$$

Similarly, the concentration of all chemical elements i within $SS_1$ can be calculated as the average of the two corresponding mirror compositions determined on the left- $(X_{k+1})$ and right-hand $(X'_{k+1})$ sides from the center of the NC.

The exact same routine procedure is further implemented to determine the concentration of all chemical elements i in all subshells. For instance, the concentration $C_{X_{k+2}}^{i}$ of the chemical element i measured at position $X_{k+2}$ is the weighted concentration of element i measured at the position $X_k (\omega_{0;2}, C_{X_k}^{i})$, within the first subshell $SS_1 (\omega_{1;2}, C_{SS_1}^{i})$, and within the second subshell $(\omega_{2;2}, C_{SS_2}^{i})$ as indicated in the Supplementary Fig. 10. The weight $\omega_{0;2}$ is proportional to the corresponding thickness $d_{0;2}$ measured from the outer surface of the NC to the surface of the first subshell $SS_1$ along the electron beam propagation $X_{k+2}$. Similarly, the weight $\omega_{1;2}$ is proportional to the thickness $d_{1;2}$, which is defined as the thickness that is probed by the electron beam through the first subshell $SS_1$ at position $X_{k+2}$. $\omega_{2;2}$ is proportional to the thickness $d_{2;2}$, which is defined as the thickness that is probed by the electron beam through the second subshell $SS_2$ at position $X_{k+2}$. Consequently, the concentration of the element i in the second subshell $C_{SS_2}^{i}$ can be calculated by using the following equations:

$$C_{SS_2}^{i} = \frac{C_{X_{k+2}}^{i} - \left(\omega_{0;2} \bullet C_{X_k}^{i} + \omega_{1;2} \bullet C_{SS_1}^{i}\right)}{\omega_{2;2}} \tag{8}$$

$$C_{SS_2}^{i} = \frac{C_{X_{k+2}}^{i} - \left(\frac{2d_{0;2}}{2d_{0;2} + 2d_{1;2} + 2d_{2;2}} \bullet C_{X_k}^{i} + \frac{2d_{1;2}}{2d_{0;2} + 2d_{1;2} + 2d_{2;2}} \bullet C_{SS_1}^{i}\right)}{\frac{2d_{2;2}}{2d_{0;2} + 2d_{1;2} + 2d_{2;2}}} \tag{9}$$

$$d_{0;2} = \frac{a}{2} - (d_{1;2} + d_{2;2}) \tag{10}$$

$$d_{1;2} = \sqrt{R^2 - \left(R - \frac{3}{2} \bullet \Delta X\right)^2} - d_{2;2} \tag{11}$$

$$d_{2;2} = \sqrt{(R - \Delta X)^2 - \left(R - \frac{3}{2} \bullet \Delta X\right)^2} \qquad (12)$$

## EDX spectroscopy: elemental maps

The elemental maps and spectra were acquired using a Thermo Fisher Scientific Titan Microscope, operating at 200 kV, which is equipped with a Super-X EDX system (comprising four silicon drift detectors). The acquisition of elemental maps was performed at a current of 1 nA for 20 min to maintain a high signal-to-noise ratio with the counts of >1.2 k per second. EDX spectra were quantified using the Thermo Fisher Scientific Velox software package version 3.0. The elemental maps showing the composition of heterogeneous and homogeneous NCs were acquired for Y, Yb, F, and Ca, while carefully checking the presence of any extra elements in the EDS spectra.

## Atomic-resolution HAADF-STEM imaging

Atomic-resolution high-angle annular dark-field scanning transmission electron microscopy (HAADF-STEM) images were acquired using an aberration corrected Thermo Fischer Scientific Titan electron microscope, operating at 200 kV. A probe convergence semi-angle and a detector inner collection semi-angle of 21 mrad and 48 mrad were used, respectively. For rare-earth based core-shell materials, it is challenging to perform electron microscopy at high resolution because of their sensitivity to the electron beam. In order to reduce beam damage, additional steps are required along with using a low electron dose and short acquisition times. It was empirically found that the negative effect of electron beam irradiation could be minimized by the deposition of a graphene layer underneath the particles under investigation. Indeed, the use of a graphene layer to prevent the sputtering of the material at the exit surface of the beam was reported in the analysis of $MoS_2$ single layer[40]. Therefore, part of the solution containing the NCs was deposited on a graphene coated conventional TEM grid. The graphene grids are prepared in-house (EMAT, University of Antwerp EP4011828A1 patent pending). Such a graphene layer prevented beam damage of the investigated NCs and image quality improved significantly. Such a preparation step is mandatory when images are acquired for quantification. $1024 \times 1024$ images were acquired in HAADF-STEM mode using a dwell time of 1 μs with a beam current of 1 pA.

## Integrated differential phase contrast (iDPC)-STEM imaging

Due to the low intensity of the lighter F atoms in the HAADF-STEM images of the homogeneous and heterogeneous NCs, we further employed integrated differential phase contrast (iDPC)-STEM imaging to study the distribution of light (F) and heavy (RE) atoms simultaneously in the crystal lattice. The iDPC-STEM image correlates to the scalar electrostatic potential field of the sample and is therefore a direct phase image, assuming that it is a thin sample[41]. Thus, it allows simultaneous imaging of both light and heavy elements.

## Downshifting photoluminescence quantum yield

Photoluminescent quantum yield (PLQY) was measured in a home-made setup. A multimode laser diode (RLT980-200gs, Roithner Lasertechnik) was mounted in a thermo-stabilized mount (TCLDM9, Thorlabs Inc.) and controlled by a laser driver (ITC4001, Thorlabs). The generated laser wavelength was 977 nm. The temperature of the laser mount was tuned to avoid the possible mode hopping during a measurement. The laser beam size was characterized with a scanning slit IR beam profiler (BP209IR/M, Thorlabs). 4-sigma values were utilized to describe the diameter of the beam.

The laser beam was directed into a 15 cm integrating sphere (QE sphere, Spectralon material, Labsphere) equipped with port reducers on both the input and output ports to reduce light escape. To prevent absorption due to water vapour in the ambient air, which is extremely prominent in the SWIR region especially when using an integrating sphere, the latter was continuously purged with nitrogen gas. The solution of colloidal NCs in toluene was introduced into a fluorescence quartz cuvette (CV10Q14, Thorlabs) with a reduced volume ((1400 μL) but with an absorption path length of 10 mm along the excitation beam. The cuvette was maintained with a cuvette holder (Labsphere), allowing to place the sample in the center of the integrating sphere.

For intensity-dependent PLQY measurements, the power of the laser was attenuated by a round continuously variable metallic neutral density (ND) filter (NDC-100C-2, Thorlabs), mounted on a step motor and controlled by an Arduino board. In order to avoid possible laser induced heating of the sample, an automated beam shutter (SC10, Thorlabs) was set in front of the ND filter. No any additional focusing of the laser beam was performed for downshifting measurements. After attenuation, the power of the laser beam was probed by a glass wedge and a single-surface reflection ( ~ 4%) was measured by a power meter (PM320 with S120C head, Thorlabs) simultaneously with a spectrum acquisition by a spectrometer. The utilization of the probing wedge, in contrast to commonly utilized glass piece or coverslip, excludes interferences between two reflections from parallel glass surfaces. As the interference pattern strongly depends on the laser beam size, divergence, displacement and incident/reflection angle, it often causes irregularities in the measured power of the reflected beam even with negligible changes in the beam path or in the laser beam itself and thus, changing the ratio between probed power and the real power inside the sphere.

All necessary for PLQY measurement spectra were measured by a fiber-coupled NIR CCD spectrometer (NIRQuest512-1.7, Ocean Insight), calibrated by the manufacturer for linearity. The entire optical system that includes the integrating sphere, optical fiber, and CCD spectrometer, was calibrated for an absolute spectral irradiance under the experimental conditions (*i.e.* full darkness and nitrogen purging) with a NIST-calibrated light source (HL3Plus, Ocean Insight). Additional check of the calibration was done by an exposure of the sphere with a laser emission having known power. The integration of the area under the measured laser spectrum was then converted into a photon flux value. The difference in the calculated value from the laser power and measured by the spectrometer photon flux did not exceed 3%.

In a typical PLQY measurement, three spectra are required according to the method described by deMello et al.[42]. Spectra for i) the empty sphere, ii) the sample in the sphere and directly illuminated by the laser radiation, and iii) the sample in the sphere but indirectly illuminated by the radiation scattered from the inner surface of the sphere should be measured. However, there are several challenges in this simple procedure such as for instance, i) low absorption of the specimen, ii) low intrinsic PLQY, and iii) the bandwidth of the generated emission is much broader compared to the one from the excitation laser. All these lead to a large difference between the intensity detected at the laser and emission wavelengths. The difference can reach several orders of magnitude in favour of the laser, thus leading to a low signal-to-noise ratio within the emission range. To overcome this problem, the downshifting emission was measured with longer integration time compared to the one used for the excitation light. To prevent the saturation of the spectrometer by the narrow scattered laser radiation during long exposure, a high-quality ultrasteep long-pass filter (LP01-980RE-25, Semrock) was inserted between the sphere and the collecting optical fibre. Therefore, five spectral measurements were performed instead of the three described by deMello *et al.*: i) one spectrum for the estimation of the photon flux from the excitation radiation within the empty sphere, ii) two spectra to calculate the modification of the photon flux under excitation at the laser wavelength due to absorption of the sample being directly and indirectly excited, and iii) two additional spectra for the determination of the photon flux of generated downshifting emission when directly and indirectly exciting the sample within the sphere.

All raw (counts vs wavelength) spectra were corrected in respect to the used integration time, detected power and, most importantly, spectral irradiance calibration, resulting in a photon flux spectra (photons/s vs. wavelength). All spectra acquisition, correction factors application, shutter operation and the motorized excitation intensity attenuation was automated using an in-house developed LabVIEW Virtual Instrument. The final analysis and combination of all measurements into a single PLQY intensity dependence was done in the OriginPRO data analysis software.

According to the uncertainty analysis of the 3 M method for absolute PLQY estimation[43,44], the main factor that influences the accuracy of the PLQY estimation is the sample absorption. Therefore, PLQY measurements were performed in concentrated nanocrystal solutions to give a relatively high-absorbed fraction of the incident photons (typical values in the range of 10%–20%). Additionally, averaging over several measured excitation/emission spectra (i.e. intensity-dependent measurements) enabled to decrease the uncertainty of determination of the integrated photon flux values. Therefore, the error of the estimation of the average PLQY value in the range 50-900 mW/cm2 varies between 1-5% (relative) for all measured samples. Note that observed differences in the error values between samples is mainly due to the difference in the absorbed fraction of incident photons. It is worth mentioning that between December 2022 and March 2023, the synthesis of the most efficient composition (i.e. heterogeneous α-$NaYF_4$:Yb:Er:Ce@$CaF_2$) was repeated three times (i.e. three different syntheses performed by two different operators with a time interval of several weeks between each syntheses) leading to absolute PLQY values of 49%, 51% and 64%. The estimation of the PLQY itself is quite accurate but a variation of efficiency can be observed from batch to batch. Note that significantly lower PLQY values have never been measured for our reference composition (including when two different operators separately performed the synthesis).

### Heterogeneous NCs micelle formulation

A micelle emulsion was formulated by mixing 100 μL of heterogeneous α-$NaYF_4$:Yb:Er-$CaF_2$ NCs (10 mg/ml) with 400 μL of a 25 mg/mL lipid solution in NaCl, and sonicated with a probe sonicator.

### Macroscopic SWIR imaging setup

The macroscopic SWIR imaging setup consisted of a liquid nitrogen-cooled InGaAs detector (640 × 512 pixels, Princeton Instruments), a 50-mm focal length lens (F2.8, Schneider-Kreuznach), a 968-nm diode laser source (Lumics) in continuous wave mode, and a combination of dichroic filters to clean up the excitation light and remove it during SWIR emission detection. The InGaAs detector was operated at −190 °C and the exposure time was set to 1 s. The camera-lens combination yielded a full field of view of 85 ×68 mm at a distance of 50 cm, corresponding to 7.5 pixels per mm. The 968-nm laser was cleaned up with a 1000 nm short pass filter. In addition, an engineered diffuser (20°, Thorlabs) was used to distribute the illumination into the whole field of view; the diffused laser radiation had a mean standard deviation intensity of $100 \pm 30$ mW/cm² in the specimen area. In the detection pathway, a series of long pass filters (both 1000 nm and 1100 nm) was used to block laser stray light.

### Fluorescence signal measurements

NCs fluorescence signal measurements were performed with the macroscopic SWIR imaging setup described above, setting the camera exposure time to 100 ms and the laser power density to 37 mW/cm² in the sample area.

### In vivo SWIR fluorescence imaging and angiography

The animal procedures were conducted in conformity with the institutional guidelines. Experiments were performed on athymic nude mice, female, 5-6 weeks old. Mice were housed at 23 °C, 45–65% humidity, with a 12/12 h light/dark cycle. Mice were anesthetized with a ketamine (Bremer Pharma GmbH, 100 mg/kg) / xylazine (WDT, 10 mg/kg) solution, and had their temperature monitored and controlled with a heating pad during the measurements. Anaesthesia was induced in a 5/6-week old mouse (Charles River Laboratories, Germany), which had a tail vein catheter implanted, and was positioned in the macroscopic SWIR imaging setup with a 1450 nm long pass filter at 1000 ms (1 frame per second). After starting the acquisition, the 968-nm laser source was switched on, and was kept on for the duration of the experiment. Following a series of pre-injection frames, 200 μL of the α-$NaYF_4$:Yb:Er:Ce-$CaF_2$ micelle formulation were injected in the tail vein catheter, with continuous imaging. Using the continuous recording of the injection, we applied a form of principal component analysis for color coding of the vascular structures based on the label arrival times using the PoissonNMF ImageJ plugin[45].

### Reporting summary

Further information on research design is available in the Nature Portfolio Reporting Summary linked to this article.

## Data availability

The experimental data used in this study are available in the open repository Figshare under accession code https://doi.org/10.6084/m9.figshare.23586075.

## Code availability

The code utilized to extract chemical profiles from EDX line scans is available in the open repository Figshare (https://doi.org/10.6084/m9.figshare.23586075). The open-source StatSTEM software is available on GitHub at the following address: https://github.com/quantitativeTEM/StatSTEM.

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

## Acknowledgements

D.H. would like to thank Dominique Ectors (Bruker AXS GmbH, Karlsruhe, Germany) for assistance and discussion on the PXRD data and TOPAS evaluations. The authors would like to acknowledge the financial support provided by the Helmholtz Association via: i) the Professorial Recruitment Initiative Funding (B.S.R.); ii) the Research Field Energy – Program Materials and Technologies for the Energy Transition – Topic 1 Photovoltaics (F.A.C., D.B., E.M., B.S.R., D.H.). This project received funding from the European Union's Horizon 2020 innovation programme under grant agreement 823717. This work was supported by the European Research Council (grant 770887-PICOMETRICS to S.V.A. and Grant 815128-REALNANO to S.B.). The authors acknowledge financial support from the ResearchFoundation Flanders (FWO, Belgium) through project fundings (G.0346.21 N to S.V.A. and S.B.) and a postdoctoral grant (A.D.B.). The authors (B.A.A., O.T.B. and A.C.) acknowledge funding from the Helmholtz Zentrum München, the DFG-Emmy Noether program (BR 5355/2-1) and from the CZI Deep Tissue Imaging (DTI-0000000248). The authors (O.T.B. and D.H.) would like to thank the Helmholtz Imaging (ZT-I-PF-4-038-BENIGN).

## Author contributions

D.H. and B.S.R. designed the original research. F.A.C. and D.H. synthesized all core and homogeneous/heterogeneous core-shell samples. F.A.C. and D.H. conducted PXRD characterization. R.P. conducted the first set of STEM experiments (low-resolution HAADF-STEM imaging and EDX line scan analyses). R.P. quantified line scan data and discussed the results and interpretation with D.G. and D.H. N.J. performed the second set of STEM experiments (EDX chemical maps, iDPC-STEM imaging, atomic resolution HAADF-STEM imaging). A.D.B. quantified the atomic resolution HAADF-STEM data and discussed the results and interpretation with S.B. and S.V.A. All optical measurements (emission spectra, absolute PLQY) were conducted by D.B. and E.M. The corresponding results and interpretation were discussed with F.A.C., D.H., and B.S.R. Fluorescence imaging experiments were conducted by A.C. and B.A.A. The analysis of fluorescence imaging data were performed by B.A.A. under the supervision of A.C. and O.T.B. D.H. prepared all figures. D.H. and B.S.R. wrote the manuscript with inputs from all co-authors.

## Funding

## Competing interests

The authors declare no competing interests.

## Additional information

¹Institute of Microstructure Technology, Karlsruhe Institute of Technology, Karlsruhe, Germany. ²EMAT, University of Antwerp, Antwerp, Belgium. ³NANOlab Center of Excellence, University of Antwerp, Antwerp, Belgium. ⁴Laboratory for Electron Microscopy, Karlsruhe Institute of Technology, Karlsruhe, Germany. ⁵Helmholtz Pioneer Campus, Helmholtz Center Munich, Munich, Germany. ⁶Functional Imaging in Surgical Oncology, National Center for Tumor Diseases (NCT/UCC), Dresden, Germany. ⁷German Cancer Research Center (DKFZ), Heidelberg, Germany. ⁸Medizinische Fakultät and University Hospital Carl Gustav Carus, Technische Universität Dresden, Dresden, Germany. ⁹Helmholtz-Zentrum Dresden-Rossendorf (HZDR), Dresden, Germany. ¹⁰Light Technology Institute, Karlsruhe Institute of Technology, Karlsruhe, Germany. ✉e-mail: andriy.chmyrov@nct-dresden.de; sandra.vanaert@uantwerpen.be; bryce.richards@kit.edu; damien.hudry@kit.edu

