## [Peer Review File · Nature Communications]

Preventing cation intermixing enables 50% quantum yield
in sub-15 nm short-wave infrared-emitting rare-earth based
core-shell nanocrystalsREVIEWER COMMENTS

Reviewer #1 (Remarks to the Author):

This work presents a comprehensive study on the impact of cation intermixing on the emission quantum yield and brightness of upconversion nanoparticles. The authors compare the atomic scale organization and optical performance of core-shell structures (prepared from the same core - alpha-NaYF₄:Yb:Er without or with Ce³⁺ co-doping) combined with either homogeneous (alpha-NaYF₄) or heterogeneous (CaF₂) shell domains. The benefit of heterostructures over homogeneous ones for preventing cation intermixing is elegantly and comprehensively demonstrated. This is a noteworthy result that will certainly impact the field.

The manuscript addresses an important problem in the current research on upconversion nanoparticles for short-wave infrared (SWIR) imaging for biomedical applications. Since the pioneering works on heterogeneous lanthanide-based core-shell nanoparticles a decade ago, the topic did not attract too much attention. The research reported in the manuscript is relevant and innovative and recommend the publication of the work.

The following points require, however, further attention:

The sentence on page 12: "(...) such a behavior indicates that the energy exchange between sensitizers and activators is significantly reduced when these optically active centers are segregated into spatially distinct domains" is not supported by experimental data and is speculative. Ion-ion energy transfer calculations – not performed in the paper - are crucial to (eventually) support the assumption. Indeed, it is probable that the publication of this paper will trigger the interest of theoreticians to perform these calculations that might shed light on the relation between emission quantum yield and brightness and the location of the optically active centers into spatially distinct domains.

A short sentence can be added to the revised manuscript anticipating the strategies that can be used to optimize the concentration of the trivalent lanthanide ions in the nanoparticles while maintaining the sub-15 nm size.

The error in the emission quantum yields needs to be estimated.

Brightness is defined as the product between the absorption strength (represented by the absorption cross section in solids or absorption coefficient in solutions) and the photoluminescence quantum yield, both quantities measured at the same wavelength. The quantity called brightness in the manuscript does not follow this definition and it is recommended that the authors address this issue.

Reviewer #2 (Remarks to the Author):

The authors show a very interesting work of rare-earth-based core-shell nanocrystals in which by avoiding cation intermixing and promoting heterogeneous shell growth, short-wave infrared photoluminescence (PL) with a quantum yield (QY) of 50% at low power excitation of at 60 mW/cm² was observed. These results are very particular, as highlighted by the authors in comparison with most recent literature. The authors present a very complete structural characterization by comparing two different batches of growth in which cation intermixing was observed, or avoided (as in the sample of interest). Finally, the authors show in vivo imaging above 1450 nm. The article is definitely interesting given the very important results of quantum yield at reasonable power densities and will be of interest to the broad readership of Nature Communications. I have few questions that might be important to respond to and to work out before further proceeding.

1) The authors discuss the heterogeneous shell formation as the major reason for the enhanced PLQY. The authors also show clearly that the cations remain well centered in the core during the growth of the hetero-shell. However, the authors also mention and cite several other articles that

implement heterostructured core-shell rare-earth-based nanocrystals. I would have appreciated a more in-depth discussion of the reason why their work and their approach displays such a large improvement of the PLQY as with respect to the others. In case it is the well-maintained cation placement in the core, what did the authors do to control this? What is the key to success here? The authors might want to point this out more clearly.

2) Which is the role of the two populations of nanocrystals as outlined initially in the paper. Do they have the exact same properties? Are there also two populations of PLQY?

3) Can the authors outline more measures to improve the PLQY via further dopant engineering in their materials. E.g. what is the optimum shell thickness, dopant concentration, materials of choice etc.? Something that could serve as a guide for other researcher's future work?

Point-by-Point Response to Reviewers

We thank the reviewers for their comments on our original manuscript and their constructive criticisms that has enabled us to improve the overall quality of our manuscript. We have undertaken a reworking of the manuscript to take into account reviewers' comments. A detailed list of the adjustments we made (in black) in response to reviewers' comments (in blue) is given below. Two versions of the revised manuscript are provided. One without any markup, and one in which all modifications are shown highlighted in yellow.

Reviewer #1

(1) The sentence on page 12: "(...) such a behavior indicates that the energy exchange between sensitizers and activators is significantly reduced when these optically active centers are segregated into spatially distinct domains" is not supported by experimental data and is speculative. Ion-ion energy transfer calculations – not performed in the paper - are crucial to (eventually) support the assumption. Indeed, it is probable that the publication of this paper will trigger the interest of theoreticians to perform these calculations that might shed light on the relation between emission quantum yield and brightness and the location of the optically active centers into spatially distinct domains.

We would like to thank the reviewer for this remark. In case it was not already clear, we would like to point out that the incriminated sentence on page 12 does not refer to our experimental results, but instead to the results published by Tan and co-workers in *Nanoscale* 2018, 10, 17771-17780 (reference 36 in the original manuscript that became reference 37 in the revised manuscript). From the comment above, it seems possible that the reviewer understood that we were referring to our own manuscript. Nonetheless, reviewer's point is valid and to clarify this issue the text on page 12 of the revised manuscript was modified to make it clear. The incriminated sentence was suppressed and now reads as:

Assuming that the chemical integrity of the core and shell domains is also retained in the heterogeneous NCs reported by Tan and co-workers,³⁷ the 10-times weaker emission observed (spatial confinement instead of co-doping) at 1000 nm is a sign that the energy transfer rate between optically active centers can be modified depending on their relative spatial distribution. To validate such an assumption, additional ion-ion energy transfer calculations would need to be performed.

(2) A short sentence can be added to the revised manuscript anticipating the strategies that can be used to optimize the concentration of the trivalent lanthanide ions in the nanoparticles while maintaining the sub-15 nm size.

We would like to thank the reviewer for this suggestion, which is integrated in the revised manuscript. The corresponding parts of the manuscript (page 12 and page 13) now read as:

Page 12: *It is worth noting that further gains could yet be made (while maintaining the sub-15 nm size) via the optimization of doping concentrations and shell thickness. This is beyond the scope of the present work, given that time-consuming nature of both synthesis and atomic scale characterization. Indeed, not only series of core NCs must be synthesized by systematically varying the concentration of one optically active element (Yb³⁺, Er³⁺ or Ce³⁺) while maintaining the other two constant, but also every series should be coated by a protective shell with different thicknesses. Such a task might soon be solved with the emergence of automated synthesis laboratories coupled to artificial intelligence techniques (e.g. machine learning algorithms) in the field of inorganic NCs.*

Page 13: *Our investigations highlights the major benefit of heterostructures over homogeneous ones for preventing cation intermixing. This enables to synthesize true core-shell structures (instead of disordered alloys) to maximize their SWIR performance in a size regime that is compatible with in vivo applications.⁵² The fact that cation intermixing between different domains can be avoided in heterogeneous NCs dramatically affects the true local concentration of optically-active centers, which is not trivial to measure accurately. The direct consequence is that optimum concentrations of the sensitizer, activator, and optically-inactive elements, which were mainly determined with homogeneous NCs, might no longer be valid for their heterogeneous counterparts. Similarly, the optimum thickness of the inert outer shell to prevent luminescence surface quenching should be reassessed. This might explain why recently published heterostructures for SWIR imaging did not reach similar performances to the one reported in this work despite the utilization of heterostructures.⁵¹ The synthetic strategy that was pioneered a decade ago by Yan and co-workers,³⁵ should be seriously considered as a promising route to enhance the photoluminescence properties of RE elements in inorganic NCs. Although CaF₂ offer interesting possibilities, it represents only one combination that was tested amongst a wide range of potential combinations. Those heterostructures, which still represent less than 10% of the reported RE-based core-shell NCs, constitute an important paradigm shift for the emergence of a new generation of highly efficient SWIR emitters with well-controlled atomic scale organization. Although the optimization of the concentration of RE elements and inert shell thickness might be of interest, the authors believe that the room for improvement will very likely be marginal without exploring truly disruptive core-shell, segmented or dimer heterostructures. The latter offer a unique possibility to modify the cation and/or anion networks between the different nano-sized subdomains. This will lead to the ability to combine materials into functional RE-based architectures, which currently have no existing counterparts. Such a synergetic coupling of properties might be the key for a new (r)evolution in the field of RE photoluminescence of nanomaterials.*

(3) The error in the emission quantum yields needs to be estimated.

We would like to thank the reviewer for this remark. The following description regarding the description of the method implemented to estimate the error in the downshifting PLQY is now clarified on page 20 (Online Methods) of the revised manuscript:

According to the uncertainty analysis of the 3M method for absolute PLQY estimation (new references [F]: Review of Scientific Instruments 2014, 85, 123115 and [G]: Advanced Photonic Research 2023, 4, 2200187), the main factor

influencing the accuracy of the PLQY estimation is the sample absorption. Therefore, PLQY measurements were performed in concentrated nanocrystal solutions to give a relatively high-absorbed fraction of the incident photons (typical values in the range of 10% – 20%). Additionally, averaging over several measured excitation/emission spectra (i.e. intensity-dependent measurements) enabled to decrease the uncertainty of determination of the integrated photon flux values. Therefore, the error of the estimation of the average PLQY value in the range 50-900 mW/cm² varies between 1-5% (relative) for all measured samples. Note that observed differences in the error values between samples is mainly due to the difference in the absorbed fraction of incident photons. It is worth mentioning that between December 2022 and March 2023, the synthesis of the most efficient composition (i.e. heterogeneous α -NaYF₄:Yb:Er:Ce@CaF₂) was repeated three times (i.e. three different syntheses performed by two different operators) with a time interval of several weeks between each syntheses) leading to absolute PLQY values of 49%, 51% and 64%. The estimation of the PLQY itself is quite accurate but a variation of efficiency can be observed from batch to batch. Note that significantly lower PLQY values have never been measured for our reference composition (including when two different operators separately performed the synthesis).

(4) Brightness is defined as the product between the absorption strength (represented by the absorption cross section in solids or absorption coefficient in solutions) and the photoluminescence quantum yield, both quantities measured at the same wavelength. The quantity called brightness in the manuscript does not follow this definition and it is recommended that the authors address this issue.

We would like to thank the reviewer for this remark. In our manuscript the term brightness is used five times, specifically:

1. Page 3 lines 14-18 - introduction:

*RE-based NCs recently appeared as promising candidates for SWIR imaging, exhibiting no photobleaching, long luminescence lifetimes (e.g. for time gated imaging), large Stokes shifts, integral coverage of the SWIR region with narrow emission bandwidths, and low long-term cytotoxicity.¹²⁻¹⁴ Nevertheless, such materials suffer from relatively low PLQYs and very low absorption cross-sections, leading to poor **brightness**.^{1,15}*

2. Page 11 line 9 - caption Fig. 5:

*c, **Brightness** measurements (normalized counts) under 980 nm excitation (37 mW/cm²) for -NaYF₄:Yb:Er (black solid bar) and - NaYF₄:Yb:Er:Ce (black open bar) core NCs with homogeneous (purple bars) and heterogeneous (green bars) shell domains. The bottom panel shows images of the emission (1300 nm long pass filter) of homogeneous (c1, c3) and heterogeneous (c2, c4) core-shell NCs under 980 nm excitation (37 mW/cm²).*

3. Page 11 line 13 - caption Fig. 5:

*All **brightness** measurements were performed on colloidal toluene dispersions with a concentration of 10 mg/mL.*

4. Page 20, line 34 - Online methods, subsection title:

Brightness measurements.

5. Page 20, line 35 - Online methods

NCs **brightness** measurements were performed with the Macroscopic SWIR imaging setup described above, setting the camera exposure time to 100 ms and the laser power density to 37 mW/cm² in the sample area.

In case 1, the utilization of the term “brightness” is perfectly in agreement with reviewer’s comment and this is left unchanged. For cases 2-5, according to the suggestion of the reviewer, we replaced the term “brightness” by the expression “fluorescence signal”. The corresponding parts of the manuscript now read as:

2. Page 11 line 9 - caption Fig. 5:

*c, **Fluorescence signal** measurements (normalized counts) under 980 nm excitation (37 mW/cm²) for α -NaYF₄:Yb:Er (black solid bar) and α -NaYF₄:Yb:Er:Ce (black open bar) core NCs with homogeneous (purple bars) and heterogeneous (green bars) shell domains. The bottom panel shows images of the emission (1300 nm long pass filter) of homogeneous (c1, c3) and heterogeneous (c2, c4) core-shell NCs under 980 nm excitation (37 mW/cm²).*

3. Page 11 line 13 - caption Fig. 5:

*All **fluorescence signal** measurements were performed on colloidal toluene dispersions with a concentration of 10 mg/mL.*

4. Page 20, line 34 - Online methods, subsection title:

***Fluorescence signal** measurements.*

5. Page 20, line 35 - Online methods

*NCs **fluorescence signal** measurements were performed with the macroscopic SWIR imaging setup described above, setting the camera exposure time to 100 ms and the laser power density to 37 mW/cm² in the sample area.*

Reviewer #2

(1) The authors discuss the heterogeneous shell formation as the major reason for the enhanced PLQY. The authors also show clearly that the cations remain well centered in the core during the growth of the hetero-shell. However, the authors also mention and cite several other articles that implement heterostructured core-shell rare-earth-based nanocrystals. I would have appreciated a more in-depth discussion of the reason why their work and their approach displays such a large improvement of the PLQY as with respect to the others. In case it is the well-maintained cation placement in the core, what did the authors do to control this? What is the key to success here? The authors might want to point this out more clearly.

We would like to thank the reviewer for this suggestion, which is integrated in the revised manuscript. The corresponding parts of the manuscript (page 13) now read as:

The fact that cation intermixing between different domains can be avoided in heterogeneous NCs dramatically affects the true local concentration of optically-active centers, which is not trivial to measure accurately. The direct consequence is that optimum concentrations of the sensitizer, activator, and optically-inactive elements, which were mainly determined with homogeneous NCs, might no longer be valid for their heterogeneous counterparts. Similarly, the optimum thickness of the inert outer shell to prevent luminescence surface quenching should be reassessed. This might explain why recently published heterostructures for SWIR imaging did not reach similar performances to the one reported in this work despite the utilization of heterostructures.⁵¹ The synthetic strategy that was pioneered a decade ago by Yan and co-workers,³⁵ should be seriously considered as a promising route to enhance the photoluminescence properties of RE elements in inorganic NCs. Although calcium fluoride offer interesting possibilities, it represents only one combination that was tested amongst a wide range of potential combinations.

(2) Which is the role of the two populations of nanocrystals as outlined initially in the paper. Do they have the exact same properties? Are there also two populations of PLQY?

We would like to thank the reviewer for this remark. It is true that optical performances of RE-based NCs are affected by their size. Thus, comparing optical performances of core-shell structures synthesized from starting core NCs with different size distributions makes sense only when size is a variable that one is examining. Nevertheless, to eliminate such a potential bias in our work, all core-shell structures (homogeneous and heterogeneous) reported in the manuscript were **synthesized from the exact same starting batch of core NCs**. This means that for both the homogeneous and heterogeneous NCs, the optically active materials (*i.e.* α -NaYF₄:Yb:Er or α -NaYF₄:Yb:Er:Ce core domains) and their imperfections (*e.g.* co-existence of more than one population) or variations in optical performances (*e.g.* due to size dispersion) is identical both for the homogeneous and heterogeneous structures. Consequently, differences observed in terms of PLQY values cannot be related to the co-existence of two populations, because these populations are present when growing homogeneous or heterogeneous structures. PLQY values are obtained by performing ensemble measurements and thus, reflect the average of all emitting particles in solution. Therefore, it is impossible to get two “populations” of PLQY. Obtaining such detailed information would require performing absolute PLQY measurements on single individual particles, which is impossible.

(3) Can the authors outline more measures to improve the PLQY via further dopant engineering in their materials. E.g. what is the optimum shell thickness, dopant concentration, materials of choice etc.? Something that could serve as a guide for other researcher's future work?

We would like to thank the reviewer for this remark that is shared with comment 2 from reviewer #1. The ratios between optically active RE elements (*i.e.* Yb³⁺, Er³⁺, and Ce³⁺) were not entirely selected randomly. The Yb³⁺/Er³⁺ composition was selected after checking up to 7 different ratios. The Ce³⁺ concentration was not optimized at all and was selected based on the reported results in the literature (for instance *Nature Commun.* 2017, 8, 737; *Nature Biotechnol.* 2019, 37, 1322). We tried to determine whether Zn²⁺ doping might have an effect as reported by Dai and co-workers (*Nature Biotechnol.* 2019, 37, 1322) . To date, we did not yet manage to synthesize heterogeneous NCs co-doped with Zn²⁺ of sufficient quality and are also not convinced that the Zn²⁺ ions enter the structure. Regarding the optimization of the shell thickness, the work still have to be done. Nevertheless, as indicated to reviewer #1 (comment 2), we expect the gains to be marginal. We believe that the truly promising direction (*i.e.* from which a major breakthrough might emerge) is in synthesizing heterogeneous core-shell architectures to take advantage of the synergetic coupling of two different inorganic domains. We fully agree with the reviewer that the scientific community should start to dig deeper into novel materials combinations.

REVIEWERS' COMMENTS

Reviewer #1 (Remarks to the Author):

The authors correctly answered the questions raised by the reviewers and the paper is recommended for publication.

Reviewer #2 (Remarks to the Author):

The authors responded well to my questions. I do not have any further concerns and would be happy to see this article published in nat com soon, so it can become a source of inspiration to other scientists soon.